# PTNQ: POST-TRAINING NON-LINEAR QUANTIZATION

## ABSTRACT

Quantization is one of the leading techniques to reduce the memory usage of machine learning models. It works by approximating the weights of a model by some function with a smaller domain (e.g., replace 32-bit floats with 8-bit integers that are coefficients in some function that maps back to 32-bit floats).

Although most quantization methods approximate weights with a linear or affine function, the weights of current machine learning models often exhibit non-linear behavior at the extremities. Moreover, some studies suggest that the extremities are important for the end-to-end accuracy.

In this paper, we introduce PTNQ, a novel post-training quantization technique that approximates weights by searching through a pool of non-linear functions. We show that PTNQ provides significant advantages over affine functions, achieving similar accuracy while requiring 2 to 4 fewer bits per coefficient.

## 1 INTRODUCTION

Quantization is widely used today to reduce the cost of inference in machine learning models. While models are usually trained with 16- or 32-bit floating-point numbers, they are typically deployed with smaller data types. This has two important benefits: 1) reduces memory consumption so the model fits in smaller devices, and 2) may reduce the computation cost (depending on the quantization method).

The most straightforward quantization method consists in approximating a weight (or parts of it) using a linear or affine function. A post-training algorithm takes the weights of a model after training, computes the parameters for a linear function so it interpolates the weights in the best way, and then replaces the weights with coefficients for that function.

Figure 1 shows the values of a channel of a weight of the OPT model (Zhang et al., 2022), sorted by increasing value. It also shows the fitting of an affine function. Although previous work has established that extremities are important for some models in terms of end-to-end accuracy (Dettmers et al., 2022), linear/affine functions do not capture these values appropriately, as can be seen in the example.

In this paper, we explore using non-linear functions for quantization. For example, in Figure 1, $\mathrm{arcsinh}$ interpolates the data much better. Non-linear functions are potentially more expensive to handle at run time (e.g., to dequantize the weights), but since memory accesses are much more expensive than arithmetic operations, it makes sense to explore the tradeoff of shrinking memory consumption even if that comes with a slight increase in the number of operations.

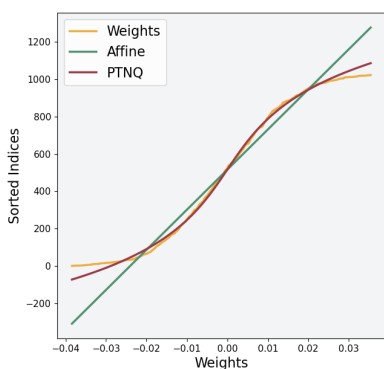

Figure 1: Sorted values of a channel of a weight of the OPT model. In blue, the best-fit affine function. In red, the data is interpolated with a non-linear function ($\mathrm{arcsinh}$).

We propose a new technique, PTNQ (post-training non-linear quantization), that searches through a pool of functions for the best fit. We focus on functions that are both invertible and differentiable so

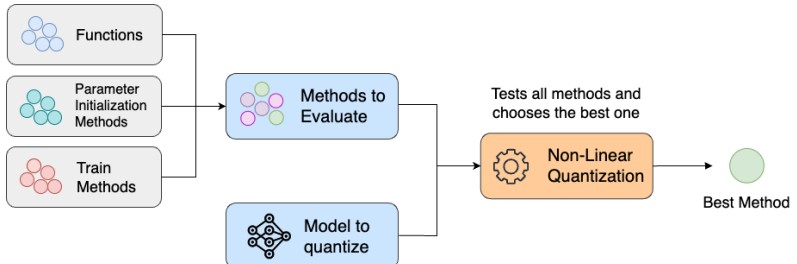

Figure 2: High-level view of the PTNQ process.

we can use the standard training machinery to find the best parameters for the functions (e.g., SGD with an LR scheduler) and so we have an easy way to dequantize the weights.

We show that PTNQ outperforms affine functions considerably. PTNQ offers similar accuracy to affine functions while requiring 2 to 4 fewer bits per coefficient. For example, PTNQ achieves the same accuracy as an 8-bit affine function with 4 to 6 bits, which is a 25-50% weight size reduction.

## 2  PTNQ: POST-TRAINING NON-LINEAR QUANTIZATION

PTNQ is a technique for quantizing weights after the training process. Weights are then dequantized during inference, and the model is executed with its original data type (e.g,, 16-bit floats).

Figure 2 gives an overview of PTNQ. At high-level, PTNQ consists of two phases:

1. Enumeration of candidate quantization methods (quantization functions and parameter selection methods).
2. Evaluation of each quantization method to select the best.

The key motivation for this pipeline is that there is no one-size-fits-all quantization method. Therefore, PTNQ searches through a list of quantization methods and selects the best one. It is often a good tradeoff to have a slower quantization process if we get to fit a model in a smaller device. The cost of quantization is largely outweigh by the cost of training and inference.

In the next sections we detail the process of generating candidates and evaluating them.

### 2.1  GENERATION OF QUANTIZATION METHODS

The first stage involves generating quantization methods. Each method has three components: a non-linear function, a method for initializing the function's parameters, and a parameter optimization algorithm. It is crucial to have a diverse pool of each of the three components so we can find the best combination for each model.

#### 2.1.1  NON-LINEAR FUNCTION GENERATION

We first assume that there is a predefined list of primitive non-linear functions. These are combined up to a user-defined depth $k$ using basic arithmetic operations (e.g., addition and multiplication) and by composing the functions with each other. The process is illustrated in Figure 3.

The set of generated functions is then filtered so that non-invertible functions are dropped. This restriction has two motivations. The first is that we need to dequantize the weights during inference. Having the inverse function guarantees that we can do it efficiently and accurately. The second is that it simplifies the optimization of the function parameters (next section).

In our implementation we use SymPy[1] to generate the inverse functions. The final set of functions and their inverses is then translated into PyTorch code, implementing both the quantization and dequantization functions.

---

[1] https://www.sympy.org/en/index.html

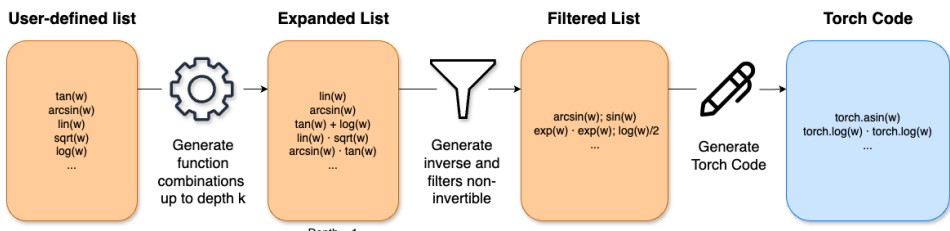

Figure 3: Function generation process. We start with a user-defined list of primitive functions. These are combined with each other up to a specified depth. The non-invertible functions are discarded. Finally, we generate PyTorch code for the final set of functions and their inverses, which is used for parameter selection and inference.

Many of the non-linear functions we use, such as logarithms and trigonometric functions, have restricted domains. Outside of their domain, they can produce undefined values that lead to instability in training and inference. To address this, when generating PyTorch code, we wrap such functions so that their input is clipped to stay within the domain. The list of all the domain guards we used is given in Table 1.

### 2.1.2 INITIALIZATION AND APPROXIMATION OF QUANTIZATION PARAMETERS

Each quantization function has a set of parameters that needs to be selected somehow. For example, for the candidate function $\cos(a \cdot x) \cdot s$, $x$ is a value of a weight to be quantized, and $a$ and $s$ are parameters. We use a different set of parameters per channel.

For each function, we try different algorithms for initializing the parameters and then approximating them. We will show later that a good initialization strategy is critical to achieve good performance. We implemented three initialization strategies: 1) initialize all parameters as ones, 2) randomly draw from a standard normal distribution with range $[-1, 1]$, and 3) a technique we call space search. The motivation behind space search stems from the observation that certain functions require initial values outside the typical range of $[-1, 1]$ supported by the other two methods.

Space search approximates parameters through an iterative process. The algorithm begins by generating random parameters with a large range of values as initial candidates. It evaluates these by calculating the MSE between the original weights and their quantized-dequantized versions, after which the top 10% are kept. From this group, the algorithm computes the average and maximum values for each channel to generate new parameters within the narrower range $[-(\mathsf{max}+\mathsf{avg}), \mathsf{max}+\mathsf{avg}]$. The whole process is repeated 50 times and, in the final step, the best parameters from the last iteration are chosen as the optimal solution.

Each function has a special, and important, scale parameter $s$. This parameter scales the output of functions to ensure they range over the full domain of the chosen quantization bitwidth. The initialization of $s$ follows the standard procedure for affine functions, i.e., it is the ratio between the input data range and the quantized output data range.

Optionally, after initialization, we can further refine the parameters by using non-linear regression. We sort the weight values and try to to fit the candidate function better by adjusting the parameters.

### 2.1.3 TRAINING QUANTIZATION PARAMETERS

Following initialization of the parameters, we then train them. The training proceess aims to minimize the layer-wise mean squared error (MSE) between the original weights and their quantized-dequantized counterparts, potentially improving the end-to-end model performance post-quantization.

We support three different training strategies, each with a different learning rate (LR) scheduler. The rationale behind focusing on LR schedulers stems from our initial setup, where the parameters are initially well-approximated. A poorly-tuned LR, especially if set too high, can destabilize the

Table 1: Domain guards for common non-linear functions. The input to these functions is clipped to stay within the allowed domain.

| Function | Domain Guard |
|---|---|
| $\log(x)$ | $x \geq 1 \times 10^{-5}$ |
| $\sqrt{x}$ | $x \geq 0.1$ |
| $\arccos(x)$ | $-0.99999 \leq x \leq 0.99999$ |
| $\arcsin(x)$ | $-0.99999 \leq x \leq 0.99999$ |
| $\tan(x)$ | Sets the input to $\pm 1$ when $x = \pm\infty$ to prevent infinite outputs. |
| $\operatorname{arctanh}(x)$ | $-0.9999 \leq x \leq 0.9999$ |
| $\operatorname{arccosh}(x)$ | $x \geq 1$ |
| $x^y$ | $x \geq 0 \vee y \geq 1$ |

training and cause parameter divergence. Therefore, it is essential to carefully adjust the LR to maintain stability and prevent the model from deviating from its favorable starting point.

The three learning rate schedulers supported are:

1. Linear learning rate scheduler, which gradually decreases the learning rate over time. Specifically, the LR of each parameter group is decayed by a small multiplicative factor linearly until a predefined milestone epoch is reached. The idea behind this approach is to provide the training process with enough initial momentum to quickly approach an optimal solution, while progressively reducing the learning rate to prevent overshooting or divergence.

2. Cosine annealing with warm restarts (Loshchilov & Hutter, 2017), in which the learning rate follows a cosine function, gradually decreasing to a minimum before restarting at a higher value. The warm restarts allow the model to escape local minima and explore the solution space more thoroughly. This technique is particularly useful in scenarios where the model might benefit from periodic boosts to the learning rate to refine the optimization process.

3. No learning rate scheduler, in which the learning rate is kept constant throughout the training process, using the initial value from the setup.

The initial learning rate was tuned to avoid disrupting the starting loss value, especially when the initial parameters were already a good approximation. The initial learning rate we use is $10^{E/2-1}$, where $E$ is exponent of the initial loss. This approach preserves good initializations by using a smaller learning rate, while allowing poor approximations to start with a higher learning rate so they converge faster.

## 2.2 EVALUATION AND SELECTION OF QUANTIZATION METHODS

The last step consists in evaluating each of the quantization methods and selecting the best one, Each method is benchmarked against commonly used domain-specific metrics, such as accuracy, perplexity, or word error rate (WER), depending on the task.

We test quantization methods by starting with the highest bitwidth (e.g., 8 bits) and reducing it one-by-one. To reduce the resources used by the process, PTNQ employs a pruning strategy: if a quantization method fails at a higher bitwidth (e.g., achieving zero accuracy), it is not evaluated again at lower bitwidths, as it is unlikely to yield good results.

## 3 EVALUATION

The goal of this evaluation is to assess whether non-linear functions offer a tangible benefit when compared with the standard affine functions for quantization. To that end, we implemented a post-training algorithm on top of PyTorch 2.2.2 and instantiated it with both affine and non-linear functions so we have the same setup for both kinds of functions. To ensure that our implementation of

Table 2: Models used for evaluation, their number of parameters, and memory required to hold the weights with 32-bit floats (the baseline). For each model, we also indicate the GPU used for the experiments, as well as their memory capacity.

| Model | Parameters | Memory | GPU | vRAM |
|---|---|---|---|---|
| ViT (Dosovitskiy et al., 2021) | 307.0M | 1.23 GB | Nvidia RTX 3070 | 8 GB |
| Wav2Vec (Schneider et al., 2019) | 317.0M | 1.27 GB | Nvidia RTX 3070 | 8 GB |
| OPT (Zhang et al., 2022) | 350.0M | 1.4 GB | Nvidia RTX 3070 | 8 GB |
| TinyLLama (Zhang et al., 2024) | 1.1B | 4.40 GB | Nvidia RTX 3090 | 24 GB |
| Phi-2 (et al., 2023) | 2.7B | 10.80 GB | Nvidia RTX A6000 | 48 GB |
| Llama3 et al. (2024) | 8.0B | 32.00 GB | Nvidia A100 | 80 GB |

affine functions is reasonable, we benchmark torchao[2] (version 0.3.1) as well, which is the official PyTorch quantization package. Both our implementation and torchao do quantization per channel.

For 8-bit integer quantization, torchao uses the standard affine technique. However, for 4-bit integer quantization, it uses piecewise affine functions (Shen et al., 2020), and thus is is potentially more precise than if using a single affine function, at the expense of increased memory consumption, since it requires more parameters per channel. We use the default group size (128), which regulates the granularity of the piecewise function.

We evaluate PTNQ with several state-of-the-art models, as listed in Table 2. We used the smallest GPU for each model to save costs.

## 3.1 QUANTIZATION FUNCTIONS

Below is the list of the 19 functions we considered for quantization, where $a$ and $s$ (scale) are learned parameters. We do not consider combinations of these functions (i.e., $k = 0$).

| | | | | |
|---|---|---|---|---|
| $x \cdot s$ | $\log(x \cdot a) \cdot s$ | $\cos(x \cdot a) \cdot s$ | $\tanh(x \cdot a) \cdot s$ | $\text{arcsinh}(x \cdot a) \cdot s$ |
| $x^2 \cdot s$ | $\sqrt{x \cdot a} \cdot s$ | $\tan(x \cdot a) \cdot s$ | $\arcsin(x \cdot a) \cdot s$ | $\text{arccosh}(x \cdot a) \cdot s$ |
| $x^3 \cdot s$ | $\sqrt[3]{x \cdot a} \cdot s$ | $\sinh(x \cdot a) \cdot s$ | $\arccos(x \cdot a) \cdot s$ | $\text{arctanh}(x \cdot a) \cdot s$ |
| $e^{x \cdot a} \cdot s$ | $\sin(x \cdot a) \cdot s$ | $\cosh(x \cdot a) \cdot s$ | $\arctan(x \cdot a) \cdot s$ | |

## 3.2 END-TO-END ACCURACY

Table 3 shows the end-to-end accuracy/perplexity/WER (as appropriate) for each of the models and quantization techniques. We observe that PTNQ outperforms affine in every case. This is expected since we include an affine function in the pool, which will be picked in the cases where it is the best option. When compared with torchao, PTNQ outperforms in all but two cases, while having less parameters and thus requiring less memory.

The key result is that we obtain an accuracy similar to baseline (FP32 weights) with only 4 to 6 bits. In contrast, affine functions need 6 to 8 bits to achieve the same results, except for the ViT model, where affine performs reasonably well with 4 bits.

In summary, PTNQ enables a 25% reduction in weight memory usage on average.

## 3.3 SELECTED FUNCTIONS FOR QUANTIZATION

We used a pool of 19 functions for the experiments, from which PTNQ selects the best function for the whole model. The parameters of the function ($a$, $s$) are selected per channel.

We now investigate how many of these functions are used in practice and whether the usage varies across models. Figure 4 shows the distribution of the functions selected for quantization for two models (across all bitwidths), two bitwidths (across all models), and overall. We show the results

---

[2]https://github.com/pytorch/ao

Table 3: Perplexity[*]/Accuracy[†]/WER[‡] for each model per method and bitwidth.

| Model | Baseline | Method | Bits 4 | 5 | 6 | 7 | 8 |
|-------|----------|--------|--------|---|---|---|---|
| Llama3[*] | 5.223 | PTNQ | 5.833 | **5.181** | **5.163** | **5.205** | **5.202** |
|  |  | affine | 6.163 | 5.449 | 5.193 | 5.249 | 5.208 |
|  |  | torchao | **5.358** | – | – | – | 5.208 |
| OPT[*] | 7.935 | PTNQ | **8.197** | **8.017** | **7.770** | **7.886** | **7.872** |
|  |  | affine | 8.903 | 8.223 | 7.806 | 7.905 | 7.916 |
|  |  | torchao | 8.768 | – | – | – | 7.917 |
| Phi2[*] | 5.910 | PTNQ | **5.778** | **5.710** | **5.761** | **5.723** | **5.694** |
|  |  | affine | 6.484 | 5.969 | 5.950 | 5.903 | 5.907 |
|  |  | torchao | 5.995 | – | – | – | 5.907 |
| TinyLlama[*] | 6.596 | PTNQ | 7.418 | **6.898** | **6.469** | **6.653** | **6.564** |
|  |  | affine | 7.655 | 6.966 | 6.502 | 6.663 | 6.605 |
|  |  | torchao | **7.043** | – | – | – | 6.606 |
| ViT[†] | 0.801 | PTNQ | **0.801** | **0.803** | **0.806** | **0.805** | **0.804** |
|  |  | affine | 0.789 | 0.799 | 0.805 | 0.800 | 0.802 |
|  |  | torchao | 0.795 | – | – | – | 0.802 |
| wav2vec[‡] | 0.02395 | PTNQ | **0.02418** | **0.02327** | **0.02328** | **0.02317** | **0.02290** |
|  |  | affine | 0.02689 | 0.02373 | 0.02340 | 0.02340 | 0.02397 |
|  |  | torchao | crash | – | – | – | 0.02397 |

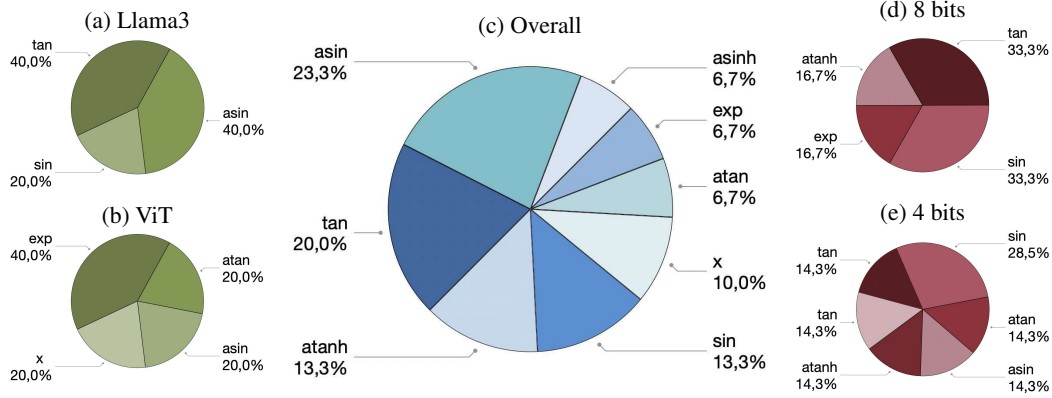

Figure 4: Distribution of the functions selected for quantization for particular models (4 to 8 bits), bitwidths (across all models), and overall.

for the best initialization mode for each combination of model and bitwidth only. Each model was quantized from 4 to 8 bits.

The first thing we note is that only 8 functions are used out of the 19 in the pool. This includes the affine function, but which is used only for 10% of the cases, showing that the traditionally-used affine functions are usually not the best choice.

Another interesting observation is that the set of selected functions is very different across models and bitwidths. For example, for the ViT model we use a different function per bitwidth, repeating only once. This means that it makes sense to instantiate PTNQ with a large pool of functions so it can select the best for each case since there is no one-size-fits-all function.

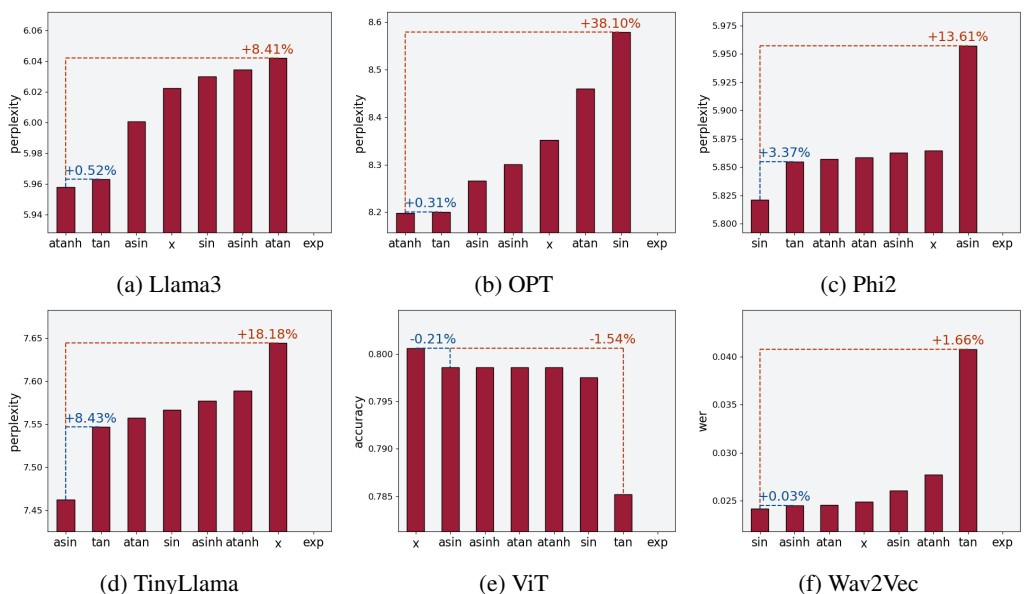

Figure 5: Comparison of the accuracy/perplexity/WER for 4-bit PTNQ between the best 8 functions. In blue, we mark the % of improvement between the best and second best functions. In red, we mark the improvement against the worst function.

We now investigate what is the contribution of each of the 8 functions to the quantization performance. For example, is the accuracy between non-linear functions similar and thus we can shrink the pool of functions? What is the additional accuracy that each function brings?

Figure 5 shows the comparison between the 8 functions. The results refer to PTNQ with 4 bits, and to the best initialization method for each function. TinyLlama is very interesting: the best function (arcsin) is over 8% better than the second best. On the other hand, arcsin is almost 14% worse than the best function for Phi2. We conclude that each function in the pool can have a substantial impact in the accuracy, and therefore the pool should be kept as large as possible.

We note that the exponential function did not yield meaningful results for 4 bits, as witnessed by the missing bars in Figure 5. However, it was the best function for almost 7% of the cases of higher bitwidths (Figure 4).

### 3.4 Impact of Initialization and Training Alternatives

We now investigate the impact of the initialization method on the accuracy, as well as whether the training step is beneficial. Figure 6 summarizes the results for 4-bit PTNQ. The 'nlr' column corresponds to the best of the three initialization methods followed by an approximation using non-linear regression.

In general, we observe that the training step is beneficial, with just one exception (TinyLlama). For the initialization method, all the four are the best for some model. Differences in accuracy between different initialization methods are substantial, which suggests that trying multiple methods is a good strategy.

### 3.5 Inference Performance and Memory Usage

Table 4 shows the time and memory usage for inference per query (batch size of 1). For PTNQ, we only show one set of numbers since they are the same for all bitwidths. This is because our prototype uses 8-bit integers always (setting the leftover bits to zero). Obviously, a production-ready implementation would pack the bits more tightly and would likely use custom kernels.

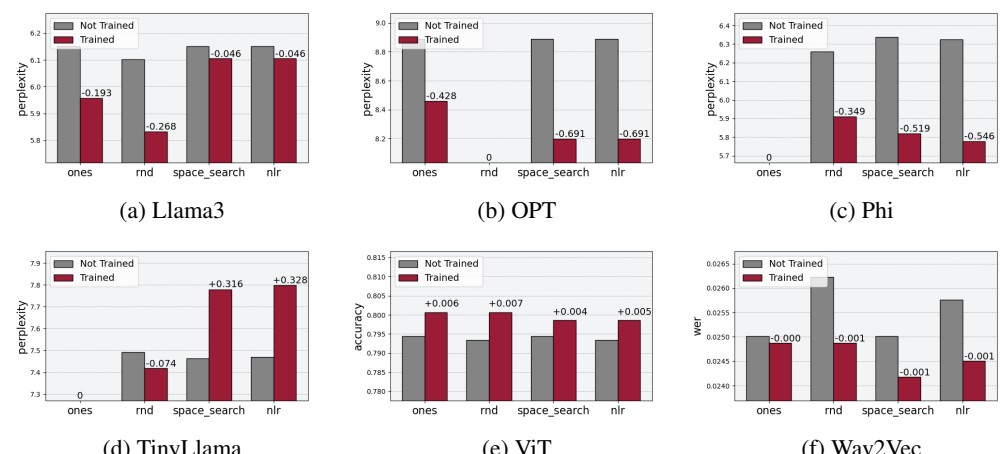

Figure 6: Comparison of the accuracy/perplexity/WER for 4-bit PTNQ for different initialization methods with and without training.

Table 4: Inference time (ms) and memory (GB) for each model per quantization method with a batch size of 1.

| Model | PTNQ | | Affine | | torchao 4-bit | | torchao 8-bit | | Baseline (FP32) | |
|---|---|---|---|---|---|---|---|---|---|---|
| | Time | Mem | Time | Mem | Time | Mem | Time | Mem | Time | Mem |
| Llama3 | 144 | 13.5 | 96 | 13.9 | 39 | 5.4 | 52 | 11.8 | 27 | 32.2 |
| OPT | 45 | 0.6 | 21 | 0.7 | 17 | 0.3 | 15 | 0.6 | 10 | 1.3 |
| Phi2 | 105 | 4.2 | 62 | 4.3 | 30 | 1.9 | 29 | 3.8 | 19 | 11.1 |
| TinyLlama | 41 | 1.8 | 20 | 1.9 | 22 | 0.7 | 19 | 1.6 | 15 | 4.4 |
| ViT | 40 | 0.4 | 19 | 0.4 | 13 | 0.2 | 16 | 0.3 | 10 | 1.2 |
| wav2vec | 36 | 0.6 | 21 | 0.6 | crash | – | 20 | 0.6 | 18 | 1.5 |

This data allow us to compare the performance between affine and non-linear functions, as well as to extrapolate the possible memory savings by comparing with torchao (which packs two 4-bit coefficients per byte).

First, we observe that PTNQ is slower than using affine functions. Since we use functions that are less commonly used in ML models, this is to be expected. Frameworks, compilers, and even possibly the hardware, may need changes to bring the performance of PTNQ on par with affine functions.

Secondly, we note that the memory consumption of PTNQ and affine functions is roughly the same, despite PTNQ having better accuracy. On the other hand, torchao is obviously faster and consumes less memory since it has a production-quality implementation. We include the results for torchao as our implementation of affine functions is directly comparable with torchao 8-bits, thus enabling the extrapolation of results for PTNQ.

## 3.6 QUANTIZATION TIME

Figure 7 shows the average quantization time (in minutes) for each model for a single quantization method (function and parameter initialization method). We further break down the time into the three main steps of PTNQ: initialization, approximation, and training.

Note that the figure shows the time for a single run of each model. In our experiments, we tested 19 functions, 4 initialization methods, 3 training methods, and 5 bitwidths yielding a total of 1140 runs per model. Nevertheless, these trials can all be run in parallel.

To reduce the used resources, we implemented an heuristic to prune early functions that showed no promise of working. Overall, 691 (10.1%) of the trial runs were cut short. Trigonometric functions

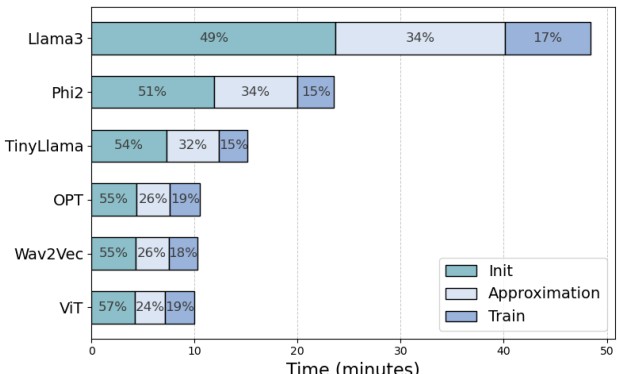

Figure 7: Quantization time per model and time per quantization step. Initialization of of p0 values; Approximation methods such as NLR; Training time. This is an average value over multiple runs

had the lowest pruning rate (as low as 1.4%). The functions that exceed 10% were: log, $x^2$, $x^3$, and $\sqrt[3]{x}$. Perhaps unsurprisingly, as can be seen in Figure 4, these functions were never selected.

## 4 RELATED WORK

The field of machine learning model quantization is vast, encompassing a wide array of techniques. For a comprehensive overview, we direct readers to recent surveys (Gholami et al., 2022; Rokh et al., 2023; Li et al., 2024) that provide a thorough exploration of the broader landscape. Here, we focus specifically on non-linear post-training quantization, addressing limitations of traditional methods in handling complex distributions.

Earlier approaches, such as those proposed by Dettmers et al. (2022); Kim et al. (2024b), attempted to handle outliers by isolating them and processing them separately using sparse algebra. However, more recent research has shifted towards methods that aim to reduce the impact of outliers without the need for separate processing. Techniques like QuaRot (Ashkboos et al., 2024) and Spin-Quant (Liu et al., 2024) propose rotating matrices to effectively eliminate outliers. While these methods show promise in maintaining model accuracy, they come at the cost of increased computational overhead due to the additional transformations required during inference.

Lookup table (LUT) quantization has emerged as a flexible alternative to the uniform quantization techniques, offering the ability to map intervals to arbitrary values. This approach, explored by Wang et al. (2022) and further developed by FLUTE (Guo et al., 2024), allows for better preservation of outlier information compared to uniform quantization. LUTs can potentially capture the distribution of weights more accurately, leading to improved model performance. However, the learning process for these lookup tables is challenging since LUTs are not differentiable, thus requiring sophisticated optimization techniques. Additionally, models quantized with LUTs consume more memory when compared to PTNQ due to the need to store the whole function mapping.

In this work we focus on the quantization of linear layers, which differs from the quantization of non-linear layers such as Softmax, GELU, or LayerNorm, which has also been explored in recent years. For example, Kim et al. (2024a) propose an approach that leverages layer-wise sensitivity analysis through SQNR to determine the optimal quantization method for each of these layers from a predefined pool of sub methods (Lin et al., 2022; Kim et al., 2021; Li & Gu, 2023).

Quantization with non-linear functions remains relatively unexplored, with logarithmic representations being the primary focus. Miyashita et al. (2016); Cai et al. (2018) proposed a base-two logarithmic quantization method for low-resolution representation of weights and activations. Building on this concept, Yan et al. (2024) introduced an integer-only scalar Power-of-Two (PoT) quantization scheme that quantizes both weights and activations into PoT representations at low precision. In this paper, we show that using a broad pool of functions provides increased accuracy and performs better in a wider range of scenarios.

Jiang et al. (2024) explore efficient hardware implementations in FPGAs for logarithm-based quantization functions. We believe similar techniques can be be used to optimize inference with the majority of the functions used in our pool.

## 5  CONCLUSION AND FUTURE WORK

We presented PTNQ, a post-training quantization technique that searches for the best non-linear quantization function from a pool. Our results show that PTNQ achieves similar accuracy to affine functions while using up to 50% fewer bits.

For future work, we plan to explore the use of different functions per layer or even per channel and assess the impact on accuracy and quantization time. Additionally, we will investigate other non-linear functions, such as more complex polynomials and combinations of functions from our test pool. We also aim to develop custom kernels to accelerate inference and study whether further hardware extensions could enhance inference performance.

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
