# OpenReview forum: "PTNQ: Post-Training Non-Linear Quantization"
_ICLR.cc/2025/Conference — Submitted to ICLR 2025_

### Official Review · Reviewer_hGib · 2024-11-01

**Soundness:** 2
**Presentation:** 2
**Contribution:** 2
**Rating:** 3
**Confidence:** 3

**Summary:**

This paper introduces PTNQ, a novel quantization technique designed to reduce memory usage in machine learning models by utilizing non-linear functions rather than traditional linear or affine methods. It highlights the trade-offs of using non-linear functions over standard affine functions, showing a reduction in bits required without significant accuracy loss. This approach enables memory-efficient model deployment without compromising accuracy, making it particularly relevant for resource-constrained environments

**Strengths:**

1. PTNQ innovates by leveraging a pool of non-linear functions, allowing for more accurate weight approximation in neural networks while using fewer bits per coefficient.
2. PTNQ’s two-phase approach first generates and evaluates various non-linear quantization functions, then selects the optimal one.
3. PTNQ explores various initialization methods, learning rate schedulers, and function combinations, providing insights into the optimal settings for different models.

**Weaknesses:**

1. I think it is unfair to compare PTNQ with affine and torchao, as both are uniform quantization methods. A fairer comparison would involve other non-uniform quantization techniques.
2. In Table 4, the inference time for PTNQ increases significantly, while the memory savings and performance improvements appear minimal.
3. This method relies on multiple steps and various heuristic combinations to determine the optimal solution, which may limit its practicality for real-world applications.
4. In terms of academic writing, there is space for improvement in the paper’s logical flow and structural clarity.
5. The innovative aspects of this work seem somewhat limited and may not yet meet the competitive standards expected for ICLR.

**Questions:**

1. Which kind of specific quantization method do you use? Like GTPQ, AWQ?
2. What is the affine function, like f(x)=x? Can you give an example? As I understand, affine function is applied in uniform quantization and non-linear function is more suited in non-uniform quantization.
3. Which model specifically did you use in the experimental part? Like llama3-8b?

---

> ### Author Response · Authors · 2024-11-14
>
> Thank you for your feedback!
>
> Below we answer your questions:
> 1) Which kind of specific quantization method do you use? Like GTPQ, AWQ?
>
> We used the more traditional quantization method of Q(w) = clip(round(w / s + z)), where w is the weight, s the scale and z the zero point. The rounding function, rounds to the nearest integer and the clip function, clips the results to the interval supported by the bit-width the model is being quantized to (i.e. for 8 bits, the interval is [-128, 127].
>
> 2) What is the affine function, like f(x)=x? Can you give an example? As I understand, affine function is applied in uniform quantization and non-linear function is more suited in non-uniform quantization.
>
> The affine function used is the usual Q(w) = w / s + z, where w is the weight, s the scale, z the zero point and the output being the quantized tensor
> The goal of our work was to use relatively simple non-linear functions to quantize linear layers, non-uniformly, since most linear layers present distributions with outliers that require a careful representation not usually present in uniform quantization
> In uniform quantization, all intervals have the same “width” and thus, the same importance to the quantization. This is the usual approach in other methods however, we aim to show that, through a small pipeline and the use of non-linear function we can achieve a better noise reduction
>
> 3)  Which model specifically did you use in the experimental part? Like llama3-8b?
>
> The models used in the experimental part as well as their number of parameters can be seen in Table 2. More specifically, the models were downloaded from HuggingFace:
>  * Vit: https://huggingface.co/google/vit-large-patch16-224
>  * Wav2Vec: https://huggingface.co/facebook/wav2vec2-large-960h-lv60-self
>  * OPT: https://huggingface.co/facebook/opt-350m
>  * TinyLlama: https://huggingface.co/TinyLlama/TinyLlama-1.1B-intermediate-step-1431k-3T
>  * Phi2: https://huggingface.co/microsoft/phi-2
>  * Llama3: https://huggingface.co/meta-llama/Meta-Llama-3-8B
>
> ----------------
> Additionally we note the following:
> torchao also does non-uniform quantization using several affine functions (128 functions for 4-bit quantization). This means the memory usage of torchao is higher than our technique that uses a single function per channel.
> The memory and performance improvements are not fully realized in our prototype because we don't do sub-byte packing of data. While torchao packs 2x 4-bit values per byte, we pack just one. This would require a more production-ready implementation, which we believe to be beyond the scope of this paper. We wanted to determine whether the community should be looking into other functions besides affine and logarithms, and we believe we successfully showed that certain trigonometric functions show great potential.
>
> A final note is that while current hardware has acceleration for some trigonometric functions, CUDA kernels used by PyTorch are certainly not optimized to handle arcsinh and similar things, since they are not used by models currently. A production implementation would fix those issues with a moderate amount of engineering work, but beyond what a small academic group can achieve.
>
> Thank you!

---

> > ### Comment · Reviewer_hGib · 2024-11-26
> >
> > Thanks Author for the detailed response. While some of my concerns have been addressed, I believe the work still falls short of the bar of ICLR. Therefore, I will retain my current score.

---

### Official Review · Reviewer_kCGD · 2024-11-03

**Soundness:** 2
**Presentation:** 2
**Contribution:** 1
**Rating:** 3
**Confidence:** 5

**Summary:**

In this paper, authors propose the PTNQ algorithm. In PTNQ, users can pre-define a list of non-linear quantization function and PTNQ would provide the best non-linear quantization function, de-quantization function and best parameters. In their experiments, they claim that PTNQ provides significant advantages over affine functions, achieving similar accuracy while requiring 2 to 4 fewer bits per coefficient.

**Strengths:**

Nonlinear quantization is a popular topic, especially when the data distribution in the tensor is not uniform. Nonlinear quantization often makes better use of resources and reduces the noise caused by quantization.

The authors provide a huge of experiments to estimate their methods.

**Weaknesses:**

This paper faces many fatal problems:

1.The motivation is not strong.
As mentioned in the article, the  significance of quantitation methods is to reduce both storage costs and computation time. The reduction in computation time depends on the increase in bandwidth benefits when data is loaded into different storage device after storage reduction. These are two goals to be achieved at the same time.

The article unilaterally emphasizes the benefits of storage, which is untenable for nonlinear quantization raise the computation time dramatically. In practice, storage is a key point, but there are more effective solutions than quantitative methods to solve the problem of purity storage problem. For example, in the advertising recommendation business, the embedding layer often uses 7z compression method for storage, and uses GPU for decompression after loading into the GPU memory. Therefore, the motivation in the article is untenable

The experiments in this paper also show this point. In table 4, the inference time of PTNQ is much larger than traditional linear quantization, but compared with linear quantization, the model size is not small significantly.

2.The method is trivial and writing is poor.

The methods in this article are very trivial. A simple yet effective method is important factor to accept this paper. However, when describing the simple method, emphasis should be placed on describing other properties of the method, such as how it is effective and how it is important in real business, rather than detailing how it is initialized. Therefore, sections 2.1.1-2.1.3 of this article should be rewritten to reduce unnecessary descriptions and further analyze the effectiveness and rationality of the method. Therefore, this article has significant shortcomings in  paper writing.

**Questions:**

How to design nonlinear quantization methods that can simultaneously balance model size and computation time?

---

> ### Author Response · Authors · 2024-11-14
>
> Thank you for your feedback.
>
> We would like to clarify one point: our technique is *not* related with quantization of non-linear layers such as embedding layers. The goal of our technique is to quantize linear layers through non-linear functions, such as trigonometric functions. As far as we are aware, this is the first systematic study on using a large pool of non-linear functions.
> By using non-linear functions to quantize linear layers, it is possible to achieve a more compact representation of the weights while maintaining model accuracy.
> Linear layers often have weights that exhibit outliers, specifically at the extremities. Traditional linear quantization fails to capture these nuances, leading to either increased model size (when skipping quantization of that layer) or a loss in accuracy.
>
> We emphasize that reducing memory bandwidth is the most important thing for the short and mid-term, at least. The gap between computation cost and bandwidth cost (in terms of energy and $) keeps widening, and thus requiring a few extra operations while halving the memory requirements is a good tradeoff.
>
> We acknowledge some of the shortcomings in the paper writing, including that it led to the confusion of non-linear layers vs non-linear quantization functions. We will fix that.

---

> > ### Comment · Reviewer_kCGD · 2024-11-26
> >
> > thank you for your reply.
> > The weakness of this paper cannot be ignored.
> > Reducing memory bandwidth is important because it brings end to end performance improved. But using the technology in this paper, the benefit  of  reducing memory bandwidth is lost. So, I cannot agree authors' view that only reducing memory bandwidth with the cost of increasing computation cost is acceptable.
> > So, I would like to maintain my score.

---

### Official Review · Reviewer_P4dL · 2024-11-04

**Soundness:** 3
**Presentation:** 2
**Contribution:** 2
**Rating:** 5
**Confidence:** 4

**Summary:**

The paper introduces Post-Training Non-Linear Quantization (PTNQ), that uses non linear functions to approximate the weights of a trained network. The technique has three components:
1. Function selection - PTNQ evaluates a broad set of non linear functions (and their combinations up-to a user defined depth k) to find the function best suited to minimize loss.
2. Quantization parameter initialization - The authors try three different initialization strategies for the parameters of the functions, namely, initializing all parameters to 1, sampling from a standard normal distribution with range [-1,1], and space search - a technique that starts by generating parameters from a large initial range and iteratively narrows the range. The parameter ranges are optionally refined using non-linear regression.
3. Quantization parameter training - After initialization, the quantization parameters are further trained to minimize the mean square error between original weights and their quantized-dequantized counterparts. The technique leverages different learning rate schedulers to optimize performance.

**Strengths:**

1. The paper is easy to follow and describes the various components of the proposed technique well.
2. The motivation is clear, and this is a relevant problem.
3. The technique does show compression advantage, however, comparison with other state of the art techniques from literature (some of which are mentioned in the related work section) is missing, making it hard to gauge the merits of the proposed non linear quantization.

**Weaknesses:**

1. The approach increases quantization time and is slower at inference compared to linear methods.
2. The technique has been only investigated on smaller models. On LLama3, the results are not much better than affine and torchao but the time and memory required for PTNQ are both higher.
3. PTNQ requires further hardware optimizations to fully leverage its non-linear functions in production settings.
4. Comparison with state of the art PTQ and QAT techniques from literature is missing in the tables.

**Questions:**

Can you share the details of the data used for training, and how many tokens were needed to train the quantization parameters?

---

> ### Author Response · Authors · 2024-11-14
>
> We thank you for your accurate and helpful review.
>
> We used different datasets depending on the domain of the model: for language models, we used WikiText, for vision models we used ImageNet, and for audio models we used LibriSpeech.
>
> As for the number of tokens, specifically for language models, we performed 1,000 iterations with batch size of 8, each with 64 tokens, yielding a total of 512,000 training tokens per layer.
> We randomized the inputs and used a fixed seed to make the results comparable among themselves.
>
> We only compared with torchao since it works out of the box with all the models we tested and it is the official PyTorch package. Also, we were a bit constrained in terms of budget. Nevertheless, we are happy to compare against other tools if you suggest us some concrete tools & algorithms you want us to compare against.
>
> Thank you!

---

> > ### Comment · Reviewer_P4dL · 2024-11-24
> > **Rebuttal response**
> >
> > Thank you for your comment and sharing experimental details. As far as comparison with other techniques is concerned, you can look at some of the newer techniques like Spinquant (https://github.com/facebookresearch/SpinQuant), QuaRot, https://github.com/spcl/QuaRot, SmoothQuant (https://github.com/mit-han-lab/smoothquant) etc. Wider comparison will make the work stronger.
> >
> > In its current form, the paper is weak and I would like to maintain my score.

---

### Meta-Review · Area_Chair_DN1o · 2024-12-19

**Metareview:**

The paper proposes PNTQ, which utilizes a list of non-linear functions to approximate the weights of a trained network. The method selects the best function in the list, initializes the quantization parameters, then train those parameters after initialization. While the motivation of the paper is sound, the authors fail to show the effectiveness of their method in a compelling way: the computational complexity is increased without much empirical validation and did not fully compare their method with existing baselines in the literature.

The reviewers unanimously judged to reject the paper, calling for more rigorous evaluation of the method and comparison with strong state-of-the-art baselines. They agreed that the current form of the paper does not meet the standard of ICLR, yet. AC encourages the authors to carefully examine the reviews and significantly update the paper for a future venue.

**Additional Comments On Reviewer Discussion:**

Reviewer P4dL mainly pointed out that the both quantization time and inference time is slower than the simple linear methods, without significant benefit in the performance (mainly for larger model like Llama3.

Reviewer kCGD raised about the memory storage issue of the proposed method, which was partly addressed by the rebuttal for a clarification argument, but was not fully convincing since the memory bandwidth reduction ended up rising the computational cost.

---

### Decision · Program_Chairs · 2025-01-22

Reject